# Research on Emotion Recognition Method of Cerebral Blood Oxygen Signal Based on CNN-Transformer Network

**DOI:** 10.3390/s23208643

**Published:** 2023-10-23

**Authors:** Zihao Jin, Zhiming Xing, Yiran Wang, Shuqi Fang, Xiumin Gao, Xiangmei Dong

**Affiliations:** School of Optical-Electrical and Computer Engineer, University of Shanghai for Science and Technology, Shanghai 200093, China; 213330562@st.usst.edu.cn (Z.J.); 211240060@st.usst.edu.cn (Z.X.); 2135061107@st.usst.edu.cn (Y.W.) 13651742686@163.com (S.F.); gxm@usst.edu.cn (X.G.)

**Keywords:** fNIRS, emotion recognition, convolutional neural network, CNN-Transformer, attention mechanism

## Abstract

In recent years, research on emotion recognition has become more and more popular, but there are few studies on emotion recognition based on cerebral blood oxygen signals. Since the electroencephalogram (EEG) is easily disturbed by eye movement and the portability is not high, this study uses a more comfortable and convenient functional near-infrared spectroscopy (fNIRS) system to record brain signals from participants while watching three different types of video clips. During the experiment, the changes in cerebral blood oxygen concentration in the 8 channels of the prefrontal cortex of the brain were collected and analyzed. We processed and divided the collected cerebral blood oxygen data, and used multiple classifiers to realize the identification of the three emotional states of joy, neutrality, and sadness. Since the classification accuracy of the convolutional neural network (CNN) in this research is not significantly superior to that of the XGBoost algorithm, this paper proposes a CNN-Transformer network based on the characteristics of time series data to improve the classification accuracy of ternary emotions. The network first uses convolution operations to extract channel features from multi-channel time series, then the features and the output information of the fully connected layer are input to the Transformer netork structure, and its multi-head attention mechanism is used to focus on different channel domain information, which has better spatiality. The experimental results show that the CNN-Transformer network can achieve 86.7% classification accuracy for ternary emotions, which is about 5% higher than the accuracy of CNN, and this provides some help for other research in the field of emotion recognition based on time series data such as fNIRS.

## 1. Introduction

In recent years, research on the theme of affective computing has become more and more active, and emotion recognition has become more and more popular in the field of great health industry and artificial intelligence [1]. The purpose of emotion recognition is to detect the emotional state of people in a specific scenario under the conditions of relevant data records. Previous research on cognitive psychology has shown that the emotional information of human emotional states is intrinsically linked to the activity of the cerebral cortex, and emotional changes can directly cause the response of the electroencephalogram (EEG) [2]. However, there are certain shortcomings in the recording of EEG signals, which are greatly affected by eye movement interference, so they still have certain difficulties in emotion recognition [3]. With the development of portable multi-channel near-infrared spectrometers, near-infrared spectrometers are considered as cost-effective and light-weight measurement systems with better spatial resolution than electroencephalograph (EEG) devices, and they are favored by more and more practitioners [4,5]. Because fNIRS data generated in response to emotional stimuli are unique, individuals cannot mask their emotions even if they want to [6]. This key advantage of fNIRS devices has been underutilized in the existing literature in the fNIRS field. Due to the poor generalization of features among participants, emotion recognition has become a difficult problem for researchers for a long time [7,8]. Our work focuses on monitoring changes in blood oxygen concentration in the prefrontal cortex (PFC) of the brain using a near-infrared spectroscopy (fNIRS) system, and mainly records the trend of oxygenated hemoglobin (HbO_2_) and deoxygenated hemoglobin (Hb) over time. Then, we use the deep learning algorithm to identify the human emotional state and obtain the inner connection between human emotional information and cerebral cortex activity.

Emotion recognition based on fNIRS technology has been studied, and there is evidence that it can be used to distinguish different emotions [9,10,11]. Yao et al. [12] studied the difference in energy consumption of human brain activation regions and prefrontal cortex regions before and after watching 3D movies, and used the support vector machine (SVM) method to perform binary recognition of brain signals before and after watching 3D movies, and the average classification accuracy reached 95.3%. In 2020, Lian Duan et al. [13] found that it is feasible to use near-infrared spectroscopy to assess brain activation caused by guilt and shame emotions, and the overall recognition accuracy of guilt, shame, and neutral emotions was 52.50%, which proved that fNIRS technology has potential in studying the neural correlates of moral emotion. Asgher et al. [14] used machine learning (SVM) and deep learning (CNN) to classify cognitive load status and achieved the best accuracy in the range of 80–87%. In addition, fNIRS data of the prefrontal cortex of the brain are also used to study other tasks. In 2020, in the study of Park et al. [15], they extracted six signal features such as the slope and mean of oxyhemoglobin and used SVM to analyze the classification of stress and mental states. In 2021, Khan et al. [16] used a variety of machine learning algorithms, especially the XGBoost algorithm, to achieve a classification accuracy of 0.77 ± 0.06 for single-handed finger taps, providing a new research direction for fNIRS-based brain computer interface applications. Meanwhile, fNIRS-based brain computer interface (BCI) studies have shown promising results on cognitive tasks [17] and depressive state classification [18]. Most of the existing studies in the field of emotion recognition use EEG signals for analysis. Wang et al. [3] extracted three features of the power spectrum, wavelet, and linear dynamics analysis, and used a variety of dimensionality reduction methods and SVM algorithms to realize the three classifications of emotions. Many researchers have achieved the classification of different emotions on the DEAP dataset. Li et al. [19] used a support vector machine (SVM) and leave-one-out validation strategy to evaluate the recognition performance and achieved the highest average recognition accuracy on the DEAP dataset. The rate is 59.06%. In 2020, Zhong et al. [20] introduced moving average technology (MA) to smooth and filter short-term fluctuations, and designed a significant regional extraction method using the attention mechanism. The recognition accuracy in the Gamma frequency band is the highest, and it is up to 73.27%. Compared with EEG devices, fNIRS systems are less expensive and more comfortable and portable. This advantage is particularly important for the accurate acquisition of emotional signals, allowing positive emotions to be collected more naturally [21].

This paper conducts a classification study of the three emotional states of joy, neutrality, and sadness based on three types of movie clips. By using the fNIRS system to monitor changes in cerebral blood oxygen concentration, we analyzed the different activation states of the prefrontal cortex of the brain due to different emotional changes. A total of 210 samples of 4 min were collected in the experiment, and the samples were processed by data segmentation. This paper proposes a CNN-Transformer network, which applies the multi-head attention mechanism to multi-channel time series data, which can better extract spatial features, and compares multiple classifiers with the network proposed in this paper to validate the latter. The accuracy of the ternary emotion recognition task is higher. This research may contribute to a better understanding of the impact of different types of video/3D movie viewing on brain function, and is of great significance for a clearer understanding of the brain’s response to different scenes.

## 2. Materials and Methods

### 2.1. Experimental Setup and Instructions

The system consists of a 2-channel light source module, an 8-channel photodetector module, and a controller. The system uses time-division multiplexing technology to control near-infrared light of two wavelengths (735 nm and 850 nm) to achieve alternate flashing, thereby measuring the concentration changes of oxyhemoglobin (HbO_2_) and deoxygenated hemoglobin (Hb) molecules in the PFC of the brain. Figure 1 shows the specific setup of the hardware system, which consists of the cerebral blood oxygen acquisition circuit part, the light source-detector module, and the data acquisition module. Before conducting this experimental study, we verified the sensitivity, linearity, stability, hysteresis, and repeatability of the system, which all met the standards of the measuring instrument.

In near-infrared spectroscopy detection, it is found that the selection of the light source-detector distance (d) has a great influence on the detection depth. If the interval is 1 cm, the measurement of the superficial cortex can be realized, but there is more noise interference. With the increase in d, the imaging depth of the banana-shaped path will become deeper [22]. When the distance exceeds 5 cm, the signal will become very weak and mixed with unusable signals [23]. When the distance (d) of the organ is 3 cm, the hemodynamic response signal of the PFC has the best measurement effect. Figure 2 shows the distribution layout of multiple light sources and detectors used in our emotion state classification experiments. In this experiment, we place the center position of the bottom two detector distances at the FPz position of the international 10–20 system to eliminate the positional uncertainty between participants [24,25].

In order to keep the system tightly fixed, we designed a flexible fit probe distribution plate to stabilize the light source and photodetector as well as their positions, which can minimize motion artifacts due to additional contact changes. The pattern of light source and detector placement in the prefrontal cortex can be customized according to the needs of the experiment, in order to locate the observation point in the opposite cortical area.

### 2.2. Participants

The Cerebral Blood Oxygen Activity State Study recruited 23 healthy participants (13 males and 10 females, age range 24.0 ± 3.0 years old) for the experiment. All participants had no history of neurological and psychiatric disorders or any known movement disorders. Written informed consent was obtained from all participants and the protocol of this study was approved by the Institutional Review Board of University of Shanghai for Science and Technology.

### 2.3. Experimental Paradigm

First, the participants were asked to sit in a comfortable chair. Before the experiment started, we asked the participants to relax for 5 min to eliminate the hemodynamic reactions left over from previous activities, and informed the participants to remain relaxed and avoid unnecessary movement or thinking during the experiment. The researcher explained the experimental procedures to the participants and ensured that the participants completed the tasks correctly. Participants sit in comfortable chairs in front of a computer screen that displays experimental tasks. Each experiment requires 3 trials, as shown in Figure 3. After each data recording session starts, there is a pre-experiment rest time of about 2 min to obtain the baseline level of blood oxygen information, then three movie clips (happy, neutral, and sad) that cause different emotional changes are played, and we ensured that the movie clips played in each experiment are not repeated. Each movie clip is 4 min. After each movie clip is played, the participants have a 1 min rest period. The trial lasts for 16 min (960 s) for a total of 48 min.

We finally obtained the fNIRS data records of 69 trial processes of the participants. Since each trial process consists of three different movie clips, each process can be split into three samples; the duration of a single sample is 4 min, so we obtained a total of 69 viewing samples of joyful emotional videos, 69 samples of watching neutral emotional videos, and 69 samples of watching sad emotional videos. Figure 4 shows a diagram of the overall experiment and data-processing process.

## 3. Data Preprocessing and Analysis

### 3.1. Calculation of Hemoglobin Concentration Changes

Under the coordination of the controller, the system collected the optical density changes before and after the chromophore absorption in the PFC of each participant’s brain at a sampling rate of 10 Hz, and obtained the hemodynamic responses of eight channels. Each channel contains light intensity information in two dimensions of oxyhemoglobin (HbO_2_) and deoxygenated hemoglobin (Hb), and a total of 16 dimensional features of blood oxygen concentration were obtained. Light exhibits low absorption and high scattering characteristics in brain tissue, and photons are absorbed by chromophores and weakened when propagating in brain tissue [26]. The fNIRS system obtains the change in light attenuation by measuring the light intensity passing through brain tissue. Then, the changes in the concentration of oxygenated hemoglobin (HbO_2_) and deoxygenated hemoglobin (Hb) molecules were calculated according to the Modified Beer–Lambert Law [27]. The odified Beer–Lambert aw equation is given below:(1)$$\Delta c_{HbO_{2}}=\frac{\varepsilon_{Hb}^{\lambda_{1}}\frac{\Delta A^{\lambda_{2}}}{DPF(\lambda_{2})}-\varepsilon_{Hb}^{\lambda_{2}}\frac{\Delta A^{\lambda_{1}}}{DPF(\lambda_{1})}}{\left(\varepsilon_{Hb}^{\lambda_{1}}\varepsilon_{HbO_{2}}^{\lambda_{2}}-\varepsilon_{Hb}^{\lambda_{2}}\varepsilon_{HbO_{2}}^{\lambda_{1}}\right)\cdot d}$$
(2)$$\Delta c_{Hb}=\frac{\varepsilon_{HbO_{2}}^{\lambda_{1}}\frac{\Delta A^{\lambda_{2}}}{DPF(\lambda_{2})}-\varepsilon_{HbO_{2}}^{\lambda_{2}}\frac{\Delta A^{\lambda_{1}}}{DPF(\lambda_{1})}}{\left(\varepsilon_{HbO_{2}}^{\lambda_{1}}\varepsilon_{Hb}^{\lambda_{2}}-\varepsilon_{HbO_{2}}^{\lambda_{2}}\varepsilon_{Hb}^{\lambda_{1}}\right)\cdot d}$$
where ΔA represents the optical density change, d is the distance from the light source to the detector, the weight ε is the absorption coefficient of different chromophores, and DPF is the differential path factor. The two working wavelengths selected by the fNIRS system are $\lambda_{1}$ = 735 nm and $\lambda_{2}$ = 850 nm. Table 1 shows the differential path factor and the absorption coefficients corresponding to different chromophores that we selected.

### 3.2. Data Filtering and Analysis

Due to the presence of physiological and non-physiological noise in fNIRS data, such as heartbeat (1–1.5 Hz for adults), respiration (about 0.4 Hz for adults), and Mayer waves at 0.1 Hz [28], we tried to use different filters for filtering and compared the results. Finally, we used a low-pass filter with a cutoff frequency of 0.09 Hz to filter the data. This can remove the above noise while retaining the original signal information to the greatest extent, and the classification result is better.

The data we obtained are the original data of a single trial process, and the relative concentration change data are obtained after the Modified Lambert–Beer Law (MBLL) is applied. As shown in Figure 5, for each channel, we averaged the relative concentration changes of HbO_2_ (red line) and Hb (blue line) for all participants under different emotional segments and displayed the waveforms (there is no filtering here). In each channel, the horizontal axis represents the range of time, the specific process corresponds to a single experimental segment in the experimental process in the previous section, and the vertical axis represents the relative concentration change of the absorbing group. In this step, the fNIRS data of eight channels show the different activation states of different regions of the PFC of the brain, and the region selection refers to the Brodmann area [29]. Among them, the relative concentrations of chromophores in channels 1, 3, 4, 5, 6, and 8 change significantly. Taking channel 8 with the most obvious trend, the concentration of HbO_2_ increases (accompanied by the decrease in Hb concentration) during the process of watching joyful movie clips, which shows that the comedy segment caused the activation of the frontopolar prefrontal right region [30], followed by a 1 min rest segment, showing a downward trend in HbO_2_ concentration. Then, there is the process of watching a neutral movie clip. In this process, the data of the eight channels can be integrated, and it can be seen that the trend fluctuation is relatively smooth compared with the other two clips. After that, there is another 1 min rest process with the relative concentration trend decreasing. Then, there is the process of watching sad movie clips. Compared with watching joyful movies, the relative concentration of HbO_2_ also has different degrees of change. In summary, it can be concluded that watching movies that represent different emotional types can cause different activation responses in multiple regions of the PFC of the brain, and the PFC area is involved in a variety of complex emotions as well as cognitive activities.

We analyzed the eight-channel fNIRS data at the same time. During the whole experiment, the brain regions with the highest activation state were the dorsolateral prefrontal cortex, the frontopolar prefrontal cortex, etc., especially in the FPC. We can consider selecting these areas as regions of interest to facilitate our subsequent selection of channels with obvious characteristics as the input of the neural network.

### 3.3. Eight-Channel Brain Region Activation State Map

In order to more clearly and intuitively observe the differences in the brain activation status of different regions of the left and right lobes of the PFC, we used the NIRS_KIT toolbox [31] to perform a first-level analysis of all channels using MATLAB R2021a. In order to reduce the impact of individual differences on the analysis of the final results, we removed some samples with poor overall trends to avoid the negative effects of outliers and averaged the experimental data samples of the remaining participants, and the data obtained were input to the toolbox to realize the visualization of the activation state of different regions of the PFC, and the location selection refers to the Brodmann area.

Figure 6 shows the average response degree of ΔHbO_2_ of selected participants under three different emotional segments. In the joyful emotional segment, the changes were most pronounced in the left and right dorsolateral prefrontal cortex (from the perspective of the participant) and orbitofrontal cortex (left) (red dashed box), which were strongly activated. It has been shown that the generation of positive emotion is accompanied by a lower HbO_2_ concentration change response in the right side of the PFC [32]. For neutral emotional clips, the brain activity is relatively stable, and the concentration of oxyhemoglobin and deoxygenated hemoglobin may remain at a relatively stable level. However, in view of the fact that the experimental process involves watching a neutral video after a 1 min break after watching a high-activation-state joy video, there may be a continued decline in the HbO_2_ concentration during the process, and the blue area of the frontopolar prefrontal cortex appears. During the processing of sad emotions, brain activity is reduced, leading to a decrease in HbO_2_ concentration, which is particularly evident in the polar prefrontal cortex (left and right), orbitofrontal prefrontal cortex (right), dorsolateral prefrontal cortex (right), and ventrolateral prefrontal cortex (right) (yellow dashed box). Some recent studies have also found that different emotions are related to different brain regions, which supports the rationality of corresponding and differentiating various emotions and hemodynamic responses. It should be noted that the relationship between emotions and parameters such as blood oxygen concentration is complex and affected by multiple factors such as individual differences and environmental factors. Therefore, in order to express the emotional state more accurately, multiple physiological and psychological indicators need to be considered comprehensively.

## 4. Classifications

### 4.1. Dataset Division

According to the previously described fNIRS data-processing habits, this paper divides the ΔHbO_2_ and ΔHb relative concentration change data of eight channels into two matrices and connects them horizontally (as shown in Figure 7), and uses a large matrix of 9600 × 16 to represent the fNIRS data of each experiment for subsequent training. In order to improve the convergence speed of the network and the accuracy of emotion recognition, we set a split_num in the process of writing the code, which means that the 4 min segments corresponding to each emotion category are divided equally. We set the split_num to 1, 2, 4, and 8 for data segmentation (Figure 7 shows the segmentation diagram when split_num = 8), 1 means to follow the original length, and 2 means to divide the time length of each original emotional segment equally to obtain twice the sample size, and so on. Afterwards, we conducted training separately and found that the highest accuracy was obtained when split_num = 8. This may be due to the fact that if split_num takes a small value, the time spent watching the same emotional clip is too long, and it is difficult to maintain the participant’s responsiveness to the stimulus at the highest response state all the time, and the information about the fNIRS data corresponding to other labels will be obtained. Moreover, a single time window will be too long and the number of samples will be small, which makes the training process difficult. Each movie clip is 4 min in duration; if we set split_num = 8, we will obtain a time window sample of 30 s. This duration is relatively long in the field where fNIRS data can be used for training, and this can provide complete emotional information. But if the number of splits is larger, emotional information may be lost, so we set split_num = 8 to prepare for subsequent training and confirmed that this is the best choice for this data segmentation. Figure 7 is a schematic diagram of data segmentation.

### 4.2. Proposed Structures of CNN-Transformer

The network structure proposed in this paper is, first, two one-dimensional convolutional neural network layers (1DCNN), with each CNN layer followed by a batch normalization (BN) layer, and the Relu function is used for nonlinear activation. Then, there is a maximum pooling layer after each convolutional layer, followed by a fully connected layer. In order to better utilize the characteristics of the convolutional neural network and extract and learn the information of the channel domain and the spatial domain in the multi-channel time series more efficiently, this paper adds a Transformer network structure after the one-dimensional convolutional neural network (1DCNN). It is a network with a multi-head attention mechanism, which can better pay attention to the spatial characteristics of multi-channel fNIRS data to train the features extracted by CNN. Figure 8 shows the network structure diagram proposed in this paper.

#### 4.2.1. Convolutional Neural Network Module

In deep learning, the convolutional neural network is one of the most widely used and typical classifiers in both image and sequence fields [33]. The CNN mainly includes the following important components: convolutional layer, activation function, pooling layer, and fully connected layer. The convolution operation is as follows: in a 1D convolution operation, the convolution kernel slides over the input sequence, and performs element-wise multiplication and accumulation for each position to extract local features. The mathematical formula of the convolution operation is expressed as follows:(3)$$y_{i}=\sum_{j=1}^{k}x_{i-j}\cdot \omega_{j}+b$$

Among them, $y_{i}$ is the *i*–th element of the output sequence, $x_{i-j}$ is the *i*–*j*–th element of the input sequence, $\omega_{j}$ is the *j*–th weight of the convolution kernel, b is the bias item, and k is the size of the convolution kernel. After the convolution operation, we add the batch normalization layer and introduce a non-linear activation function ReLU. The mathematical expression is as follows:(4)ReLU(x)=max(0,x)

Then, we add the pooling operation, which is used to reduce the dimension of the feature map and retain the most significant features. We use the MaxPooling layer, which selects the maximum value in a certain window as the pooled value. After the above two convolutional blocks, a dense fully connected layer is added, followed by a Transformer network.

#### 4.2.2. Transformer Module

The Transformer network is a network architecture that relies on the attention mechanism. It does not have a loop structure like RNN and can use GPU for fast parallel computing. It was first proposed by Vaswani in 2017 [34]. It is mainly used to process sequence data, especially in the field of natural language processing (NLP). The Transformer network in this paper mainly includes the following structures: self-attention mechanism, multi-head attention mechanism, and feed-forward neural network layer. Given an input sequence *X*, the formula for computing the self-attention representation Z is as follows:(5)$$A\mathrm{ttention}(X)=Soft\max(\frac{XQ^{T}}{\sqrt{d_{k}}})V$$
where *Q*, *K*, and *V* are the Query, Key, and Value matrices obtained by linear transformation of the input sequence *X*, respectively; $d_{k}$ is the dimension of the query vector *Q*, and finally the attention weight is multiplied by the value matrix to obtain the self-attention representation Z. The calculation formula of the multi-head attention mechanism is as follows:(6)$$MultiHead(X)=Concat(Head_{1},Head_{2},\ldots ,Head_{n})\cdot Wo$$
where $Head_{n}$ represents the attention representation of the nth head and uses Concat to stitch multiple attention representations together; Wo is the weight matrix for linear transformation. Finally, the feed-forward neural network layer is added, which is used for nonlinear transformation and feature extraction of self-attention representations. The calculation formula of the feed-forward neural network layer is as follows:(7)FFN(Z)=ReLU(ZW1+b1)W2+b2
where $W_{1}$, $W_{2}$, $b_{1}$, $b_{2}$ are the weight and bias matrices for linear transformation; ReLU(x) is the modified linear unit activation function. The subsequent Add&Normalize layer adds the input data and the calculated features, allowing the model to learn the residual representation, which can alleviate the problem of gradient disappearance and speed up the training process of the model. Finally, the attention is turned to the high-dimensional space through the Linear layer and the result is output through the Softmax layer.

#### 4.2.3. Network Training Parameters

To better understand the structure proposed above, we give the input and output shapes of the data at each layer, and summarize the parameter information in Table 2. All hyperparameters of the proposed CNN-Transformer network are specially tuned to achieve the best accuracy and convergence speed. For the Transformer network part, we set the TransformerBlock parameters embed_dim = 32, num_heads = 2, ff_dim = 32. Finally, training is performed through the cross-entropy loss function and the SGD optimizer; the learning_rate is set to 0.005, and the batch_size is 32. In addition, since we collected 69 4 min clips for each emotion clip, there are a total of 207 long window samples. We selected the data slightly and obtained about 1600 samples after applying split_num = 8. We set test_size = 0.2, that is, 1280 samples were used for training and 320 samples were used for testing.

### 4.3. Results and Discussion

We trained the model based on tensorflow2.12.0 on GPU Nvidia RTX3070. fNIRS data are typical time series data, which corresponds to the TSC method in the field of machine learning [34]. Based on the cross-entropy loss, we trained each TSC method for deep learning for 300 epochs. We first used decision tree, random forest, support vector machine (SVM), K-Nearest Neighbor (KNN), XGBoost, and other machine learning classifiers to train the data. It is worth noting that the best performance is the XGBoost classifier, whose average accuracy can reach 80.5%. Then, we used the convolutional neural network (CNN), Long Short Term Memory network (LSTM), and Transformer network for training. The classification accuracy of the convolutional neural network can reach 82.6%. However, compared with XGBoost, the performance of the convolutional neural network on these self-built datasets is not outstanding. The CNN-Transformer network proposed in this paper produces an average recognition accuracy of 86.7%, which shows a good effect in this research field. Figure 9 shows the result of the confusion matrix of all classifiers (label 0 represents joyful emotions, label 1 represents neutral emotions, and label 2 represents sad emotions).

Overall, we noticed that machine learning and deep learning algorithms have the highest ability to identify joy emotions, which is because the HbO_2_ parameters are fully activated when watching comedy, and this activation feature is fully utilized. The recognition accuracy of neutral emotions (label 1) is the lowest, because some data have different degrees of fluctuations, which may be predicted as label 0 or label 2, while the probability of sad emotions being recognized is in the middle. Regarding this problem, it may be that the participants’ reaction to the previous video has not completely disappeared while watching the neutral emotion video, and we should select more neutral emotion clips to collect better neutral emotion data. We have tried to adjust the parameters many times and sorted out the results of each classifier for recording. In order to understand the performance of each classifier more clearly and facilitate comparison, as shown in Figure 10, we give the classification accuracy map of each classifier for ternary emotion recognition.

As can be seen from the above figure, XGBoost has an absolute advantage in machine learning algorithms, and the average recognition accuracy of CNN in deep learning algorithms is comparable to it. The CNN-Transformer network proposed in this paper uses convolution operations and attention mechanisms to make it significantly better in the ternary emotion recognition task than other classifiers, and it is stable overall. Figure 11 shows the graph of the training accuracy (accuracy) and loss (loss) during the training process of the CNN-Transformer network. It can be seen that after 500 epochs, the model gradually converges.

In order to enhance the validity of the results, this study conducted multiple results tests and used IBM SPSS statistical software to conduct in-depth statistical analysis of the results. We used the Friedman test statistical method. Table 3 shows the Friedman test analysis results. 

Through the Friedman test analysis results table, it can be seen that the CNN-Transformer algorithm has the highest median, while the SVM algorithm has the lowest median. The significance *p* value is 0.000 ***, so the statistical results are significant, indicating that there are significant differences between Decision tree, Random forest, SVM, KNN, XGBoost, CNN, LSTM, Transformer, and CNN-Transformer; the difference magnitude Cohen’s f value is 0.718, which means there is a large difference.

## 5. Conclusions

In this paper, considering that the fNIRS system is cost-effective, convenient, and comfortable, and has a higher spatial resolution of the signal, the activation and response states of the PFC of the brain were collected for 23 participants watching three different types of emotional clips, and the different brain activation regions induced by the different emotional fragmentation processes were analyzed and investigated. We collected fNIRS data from eight channels in the PFC in this work to build a dataset for ternary emotion recognition. We applied nine classifiers, comprising five simple machine learning algorithms, and three deep learning algorithms of CNN, LSTM, and Transformer, and a CNN–Transformer network, innovatively proposed to classify self-built emotion recognition datasets. Among them, the machine learning algorithm XGBoost has the best classification effect, and its average classification accuracy can reach 0.805. On the other hand, since the convolutional neural network (CNN) has an average classification accuracy of 0.82 in this task, the advantage is not obvious for other algorithms. Therefore, this study used two layers of 1D convolutional neural network processing and then input it to the Transformer network with a multi-head attention mechanism, and made appropriate parameter adjustments. After 500 epochs of training, the accuracy can reach 0.867, which is a good result in the field of emotion recognition. In this study, the data-collection and processing methods, as well as the innovative use of the CNN-Transformer network to classify the three-element emotion of the self-built dataset, which achieved good results, all provide references for the development of other related research in this field. It fills the gap in emotion recognition research using fNIRS technology. These are the preliminary results of this study. In the future, we will add more channels to the cerebral cortex, conduct more global research, and invite more participants to conduct experiments to further improve the generalization ability and emotion recognition accuracy.

## Figures and Tables

**Figure 1 sensors-23-08643-f001:**
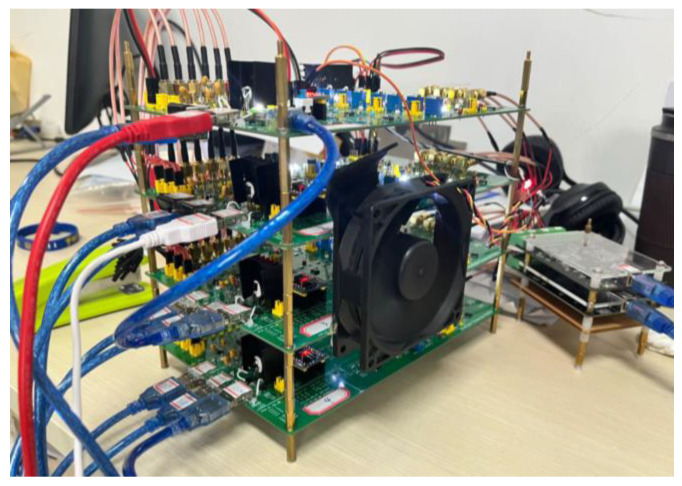
fNIRS system hardware components.

**Figure 2 sensors-23-08643-f002:**
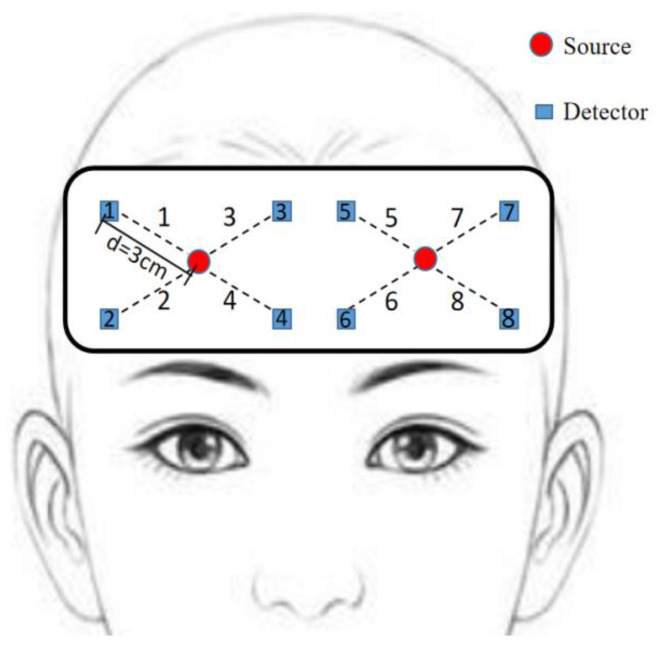
Light source and detector layout.

**Figure 3 sensors-23-08643-f003:**
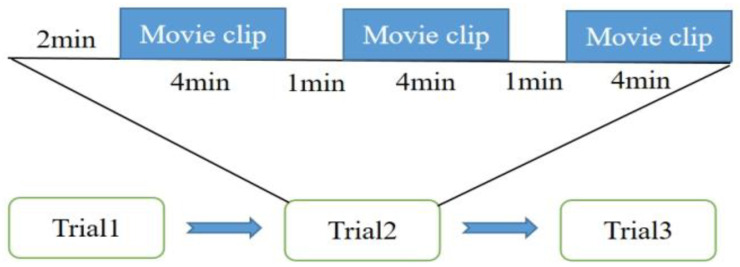
Sequence diagram of the experimental paradigm.

**Figure 4 sensors-23-08643-f004:**
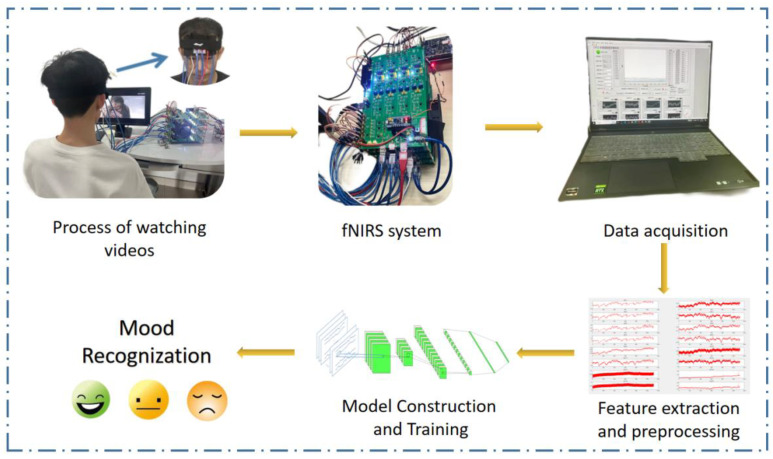
Overall experimental process and data-processing process diagram.

**Figure 5 sensors-23-08643-f005:**
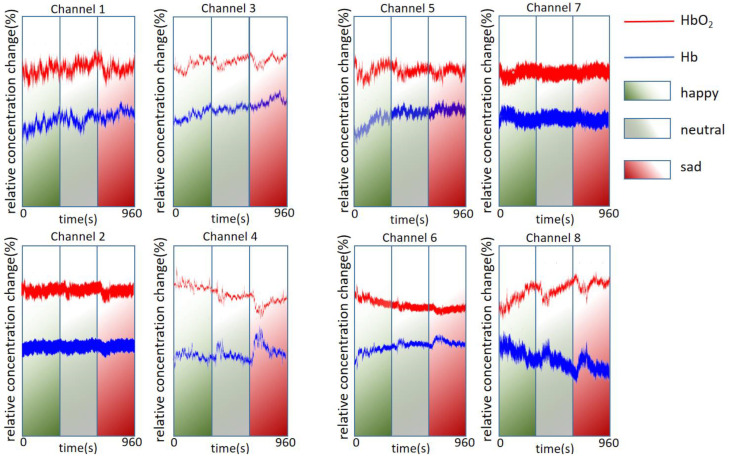
fNIRS signal waveforms for each channel. Each channel shows the average response of all participants to different emotional segments.

**Figure 6 sensors-23-08643-f006:**
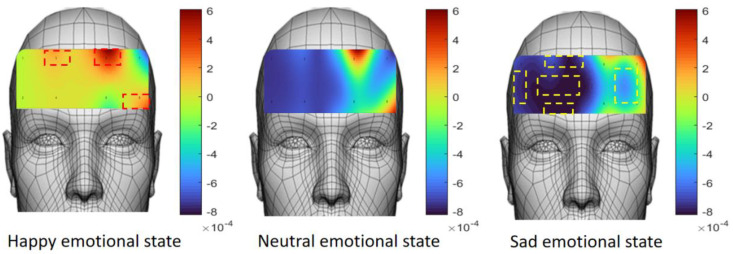
Visualization of activation states in different regions of prefrontal cortex (ΔHbO_2_).

**Figure 7 sensors-23-08643-f007:**
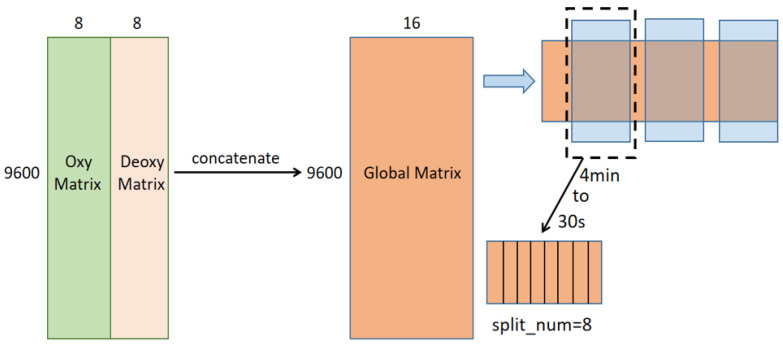
Data slicing diagram.

**Figure 8 sensors-23-08643-f008:**
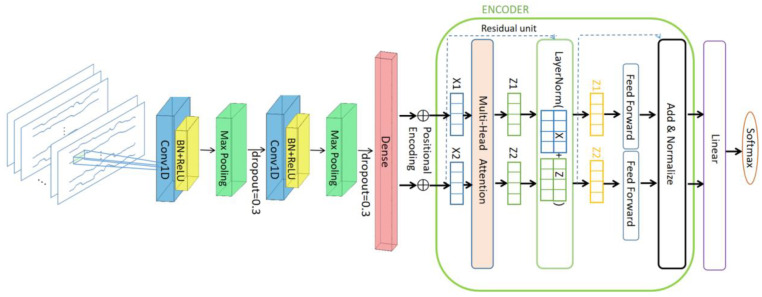
CNN-Transformer network structure.

**Figure 9 sensors-23-08643-f009:**
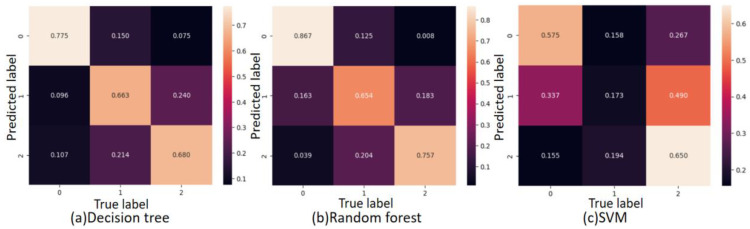
Confusion matrix for eight classifiers.

**Figure 10 sensors-23-08643-f010:**
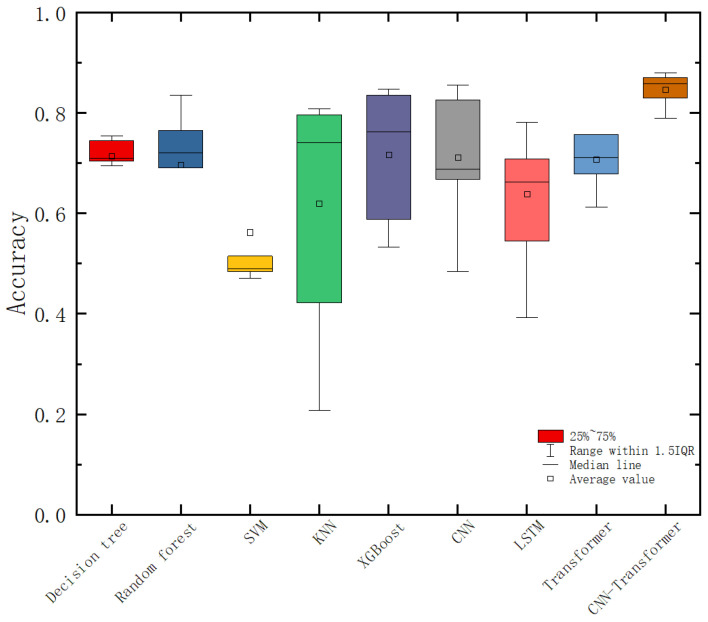
Boxplots of recognition accuracies for all classifiers.

**Figure 11 sensors-23-08643-f011:**
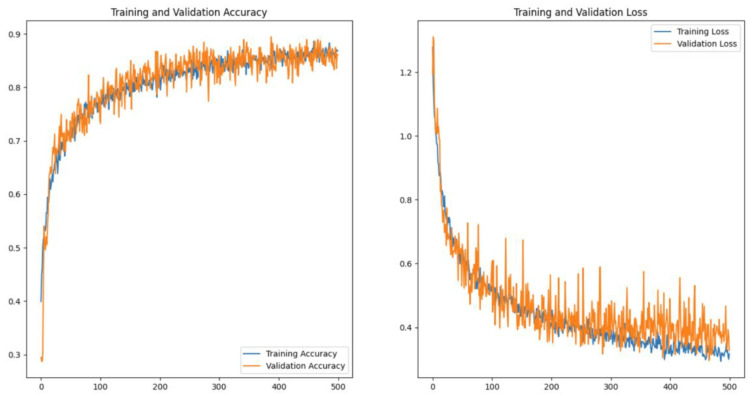
The loss and accuracy of CNN-Transformer in training process.

**Table 1 sensors-23-08643-t001:** The parameters we used for Modified Beer—Lambert Law.

Wavelength (nm)	DPF (cm)	$\varepsilon_{HbO2}$ (cm^−1^)	$\varepsilon_{Hb}$ (cm^−1^)
735	6.058	0.4646	1.2959
850	5.5	1.1596	0.7861

**Table 2 sensors-23-08643-t002:** Input and output size of CNN-Transformer.

Layer	Input Size	Output Size
InputLayer	(None,300,16)	(None,300,16)
Sequential	(None,300,16)	(None,29,32)
TransformerBlock	(None,29,32)	(None,29,32)
Flatten	(None,29,32)	(None,928)
Dropout	(None,928)	(None,928)
Dense	(None,928)	(None,500)
Dense	(None,500)	(None,3)

**Table 3 sensors-23-08643-t003:** Friedman test analysis results table.

Variable	Sample Size	Median Number	Standard Deviation	Statistics	*p*	Cohen’s f Value
Decision tree	18	0.709	0.084	55.422	0.000 ***	0.718
Random forest	18	0.72	0.105			
SVM	18	0.49	0.141			
KNN	18	0.69	0.212			
XGBoost	18	0.791	0.1			
CNN	18	0.752	0.098			
LSTM	18	0.663	0.102			
Transformer	18	0.711	0.045			
CNN-Transformer	18	0.859	0.032			

Note: ***, **, and * represent the significance levels of 1%, 5%, and 10% respectively. We analyzed whether the *p* value of each analysis item is significant (*p* < 0.05). Cohen’s f value indicates the size of the effect. The critical points for distinguishing small, medium, and large effect sizes are 0.1, 0.25, and 0.40, respectively.

## Data Availability

The data behind the results of this paper have been deposited in the laboratory database and can be obtained from the authors after the article is accepted.

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
