# Peer review of "Research on Emotion Recognition Method of Cerebral Blood Oxygen Signal Based on CNN-Transformer Network"

_sensors, 2023, doi:10.3390/s23208643_

Round 1

Reviewer 1 Report

After reading several times the submitted article, I find it to be interesting for the scientific community. Indeed, the number of articles regarding emotional detection using EEG is much bigger than that using the presented idea. Thus, the approach presented by the authors is innovative and worth studying. However, before publishing the paper there are certain points that the authors should clarify or expand the given information.

First of all, the authors have not specified the fNIR collecting device/headset that they used during the data collection. Is it a device built by the researchers or is it a commercial headset? I find this question to be relevant and that this information should be added to the article. In fact, one of the reasons for this work is that fNIR is "more comfortable and convenient" and that "near-infrared spectrometers are considered as cost-effective and light-weight measurement systems". Thus, for a case in which the experiment had to be reproduced, it would interesting for the reader to know the model and technical specifications of the equipment used for the study.

Second, with regard to section 2.1, there are two parts of the text that are practically redundant. More precisely, the second sentence of the section's first paragraph and the last sentence of the second paragraph of the section. Please, merge these sentences so that such redundancy does not occur. In addition, there is an explanation about the proper source-detector "d" distance. In my opinion, the article would improve in clarity if the authors marked distance "d" in Figure 1.

Moving onto subsection 2.3., there are several image and video databases that are considered to be standard for elicitating different emotions. Have the authors used any of these clips? For the sake of repeatability, is it possible for the reader to access those videos or at least to know if the belong to any specific film or video database? Also, did the authors consider any specific order for displaying the videos or did the show the in a random order to the participants? If the videos were always displayed in the same order this could bias the experiment towards detecting one emotion better that the others.

Apart from this question, the authors should correct their reference to Figure 3 in the last sentence of page 4, as there is no indication to those 4 minutes in that specific Figure 3.

Then, there is also a problem with the equation numbering as in many of the equations the numbering identification of the formula has moved to the left or below the equation itself. Please, realocate the numbering to the right of the formula so that it appears in the same row as the identified formula.

Concerning subsections 3.2 and 3.3., in my opinion they could be merged into a single subsection as subsection 3.2 has not much information. In this subsection the authors state that they tried two types of filtering: a low-pass filter and a band-pass filter. However, these two filters have almost the same cutoff frequencies and seein that the low pass filter performed better, the authors decided to use it instead of the band-pass one. But that's all the subsection, which is not specially relevant for the research. Thus, I suggest just leaving that the authors used the low-pass filter and that they merged together subsections 3.2. and 3.3. Also, the authors should refer to DELTA_c_Hb and DELTA_c_HbO2 in subsection 3.3, as they are talking about the previously obtained variations of the concentartion of the hemoglobin and not the absolute values.

Also, concerning Figure 4, the authors should put a title to each chart so that the charts can be linked to an specific channel of the fNIR signals. And, continuing with that figure, the authors should also mark on the chart those 1 minute intervals related to the resting period in between clip visualization. Or do those 1 minute time intervals belong to the "joy, neutral, sadness" emotions?

Another point to take into account is that certain concepts could be explained better, such as happens with the "split_num" parameter. After reading the document, the reader is capable of understanding what this parameter does. However, I could be better explained. In fact, some parts of the explanation sound contradictory: "Moreover, considering that too short slices will lead to incomplete emotional data, we set split_num=8 to prepare for subsequent training and confirmed that it is indeed the optimal choice". If I have understood correctly, setting a higher split_num value does reduce the size of the analysed windows and increases the amount of windows to be analysed. Hence... how is it possible that a shorter windows size is the optimal choice if it may have incomplete emotional data?

In addition to that, when explaining the data partition approach, the authors state that they built a 9600x16 matrix by concatenating the Oxy and Deoxy matrices horizontally and that they later split the 4 minutes corresponding to the video displaying time periods using the split_num=8 parameter. Nevertheless, they do not mention how they process the 1 minute inter-video intervals. Anyway, if the matrices have 9600 samples, which corresponds to 960 seconds at a 10Hz sampling rate, it has to be that they include the samples of the 2 minutes of relax previous to the videos as well as the 1 minute intervals in between videos. How have the researchers processed these time intervals? What kind labels did the instances of these time periods have? To me, this is an important aspect to clarify as, being 4 minutes per trial, a 25% of the samples of each trial correspond to these "relaxing/resting" intervals. Not having explained this process properly leaves certain doubts on the technical soundness of the experimental process.

Finally, my last but not least important comment concerns subsection 4.3. In my opinion there is an important gap to fill when it comes to describing how the results were obtained. In this subsection the authors stated that "Based  on the cross-entropy loss, we train each TSC method for deep learning for 300 epochs. We first used decision tree, random forest, support vector machine (SVM), K-Nearest Neighbor (KNN), XGBoost and other machine learning classifiers to train the data.". First of all, if the authors trained other classifiers they shoul list them or else, avoid mentioning that they did that as it is irrelevant information.

Besides, there is no information about the validation method they used. How much of the data was used for the training? Did the authors use any specific cross-validation approach? Also, athe article refers all the time to the training, but what about the testing? Didn't the authors keep a part of the data for the sake of testing the trained algorithms? If so, how much? In my opinion there is important information missing in this section that must be included if the paper is to be published.

To continue with the problems of subsection 4.3, the authors stated that "XGBoost has an absolute advantage in machine learning algorithms, and the average recognition accuracy of CNN in deep learning algorithms is comparable to it". After seeing Figure 9, it is true that median accuracy of the XGBoost is higher than that of the Random Forest (RF) one, for example. Nevertheless, if Figure 9 shows the distribution of the accuracies obtained over 300 runs, then the XGBoost has a big part of its distribution way below the levels of the RF. This may lead to think that the XGBoost has a high variability in its results, ranging a high part of its distribution lower than the RF. Hence, this type of assumptions done by the researchers are a bit subjective and lack technical robustness. Also, there are more metrics different to the accuracy that would give a better information on how the algorithms are performing: F-scores, precision, recall, area under the ROC curve... Moreover, in the literature there is a variety of statistical methods to farily compare algorithms one against the other, as, for example, the Bayesian Correlated t-test. Or, maybe, the authors could apply the simpler Friedmann test to, at least, confirm that the differences in performance are statistically different. The paper would greatly benefit from extending the analysis of the results, adding some more metrics or doing a more thorough algorithm comparison.

So, overaly, I consider that the idea underlying the article is interesting and innovative, but there are some important flaws that should be corrected/improved in other to consider the article suitable for publication.

Concerning the writing style of the document, I have found some issues that negatively impact the quality of the paper. First of all, there are several sentences that are excesively long or that try to express too many ideas. This makes them quite difficult to read and understand and the text would improve if they were split into shorter ones. Two examples are in page 8 (lines 45-50) and page 9 (lines 89-93).

Besides, there are some sentences that seem unfinished, badly built or not well connected to the rest of the text. These are some examples:

-" As show in Figure 7, it is the network structure...". --> Should be something like "Figure 7 shows the network structure..."

- "This may be due to the fact that if... , and information about the fNIRS data corresponding to other labels will be obtained. And it is difficult to achieve the best classification accuracy." --> That last sentence seems independent from the rest of the text.

- The first paragraph of subsection 2.3 is written in a style that resembles an instruction manual rather a description of the experimental paradigm. Besides, that whole subsection is written in present and not in past, which differs in style from the rest of the paper.

- "Convolution operation: In a 1D convolution operation, the convolution kernel... local features." --> Lacks any type of linkage to the rest of the paragraph.

- "The two working wavelengths lambda1=735nm and lambda2=850nm selected by the fNIRS system, Table 1 is the differential path factor we selected and the corresponding absorption coefficients of different chromophores."

Apart from that, the authors make a relatively inconsistent use of capital acronyms. If a term is not going to be used again in the text, then there is no need to define an acronym: DPC, FPCR, FPC, ROI... All those terms appear just once in the text and defining their acronyms has no sense. In fact, it only makes the text more difficult to understand. On the other side, the term "prefrontal cortex" is used over 20 times but its acronym PFC is only used three times. That's a clear case in which using the acronym makes sense.

Continuing with language style, the authors of the paper must revise the use of capital letters as they are very incosistent. The paper has several conflicts in this sense, here are some examples in which normal words use a non justified capital letter: "... consists of three Movie clips...", "... a total of 69 clips and 69 Samples....", "Convolution operation: In a 1D convolution operation, the convolution kernel... local features", etc. Also, the authors are not consistent when they use capitals in technical terms. The following example (which is not an isolated one) shows how the same term has been written in 3 different ways in the same sentence: "Finally, the Feed-Forward Neural Network Layer is added, Feed-forward neural network layers are used... representation. The calculation formula of the feedforwards neural network layers is as follows:...". Note that this not only applies to capitals, but to the use of the dash.

Author Response

Please see the attachment. For convenience, I have attached the revised manuscript to the Response Letter. Thanks for your hard work

Reviewer 2 Report

The paper reports about the development of a deep learning method able to provide a 3-classes classification of emotions. The topic is interesting, however, in my opinion, few concerns need to be addressed:

1) please provide some information regarding the fNIRS system employed. From figure 3, it seems that it is a home-made device. Please, specify how it was validated against a gold standard in order to strengthen the findings of the paper.

2) It is not clear to me why after a band-pass filter with cutoff frequencies of 0.01 and 0.09 Hz, a second low-pass filter with the same cut-off frequency has been applied. Please stress this aspect in the manuscript.

3) Concerning the stimulation, did you randomize the order of the emotions presented to the participants? In case, do you exclude the possibility to have a bias regarding the order of the stimulation? Please specify this aspect in the manuscript.

4) Concerning the machine learning analysis, it is not clear to me how the training and test sets have been evaluated. Did you randomize the choice of the samples? Did you test several partitions of the study sample and reported the average result? Moreover, from the same participants you obtained several temporal windows, did you put all the samples from the same participant in the same fold? If not, it could produce an overfitting effect. Please specify this aspect in the manuscript.

5) The acronyms are explained several times in the manuscript, please use the acronym once it is defined.

6) The paper is sometimes redundant. Please remove the parts when an information is repeated (e.g., the wavelengths used are reported in both line 109 and 128).

7) Please, report a statistical analysis to compare the accuracies obtained by the different machine learning approaches, in order to statistically establish whether a method outperforms the others.

8) In my opinion, the Authors could consider preserving the standard organization of papers, preserving the Results and Discussion sections. In this way the readability of the paper would increase. In this way further future directions and impacts of the research could be mentioned in the manuscript.

Author Response

Please see the attachment.For convenience, I have attached the revised manuscript to the Response Letter.Thanks for your hard word.

Reviewer 3 Report

This study explores emotion recognition using cerebral blood oxygen signals, specifically employing a functional near-infrared spectroscopy (fNIRS) system to record brain signals while participants view various video clips. The research focuses on analyzing changes in cerebral blood oxygen concentration in the prefrontal cortex and uses multiple classifiers to identify joy, neutrality, and sadness emotions. Initially, a convolutional neural network (CNN) was tested but did not significantly outperform the XGBoost algorithm. Consequently, the paper introduces a novel CNN-Transformer network designed for time series data. This network combines convolutional operations to extract channel features with a Transformer structure and multi-head attention mechanism for enhanced spatial information. The results demonstrate an 86.7% classification accuracy for ternary emotions, a 5% improvement over the CNN, offering potential benefits for emotion recognition research based on time series data like fNIRS.

Strengths:

Comprehensive Analysis: The results and discussion section provides a thorough analysis of the classification performance of various machine learning and deep learning algorithms, offering a comprehensive understanding of their strengths and weaknesses.

Clear Presentation: The section is well-structured and uses clear figures (e.g., Figure 8 and Figure 9) to visually represent the performance of different classifiers, making it easier for readers to grasp the findings.

Benchmarking: The study benchmarks multiple classifiers against each other, including XGBoost, CNN, LSTM, and the proposed CNN-Transformer network, allowing for a direct comparison of their performance in emotion recognition.

Significant Improvement: The CNN-Transformer network demonstrates a substantial improvement in accuracy (86.7%) compared to other classifiers, highlighting its effectiveness in the research field.

Convergence Analysis: The inclusion of a graph illustrating the training process of the CNN-Transformer network (Figure 10) helps validate the model's convergence, adding credibility to the results.

Weaknesses:

Limited Details on Hyperparameters: The discussion does not provide in-depth information about the hyperparameters used for training the classifiers, making it difficult for others to replicate the experiments or fine-tune the models.

Absence of Statistical Tests: While accuracy values are presented, statistical tests (e.g., significance testing) to determine if the observed differences in accuracy are statistically significant are missing. This would strengthen the validity of the findings.

Limited Insight into False Positives/Negatives: The discussion briefly mentions that neutral emotions are challenging to classify, but it does not offer insights into why certain misclassifications occur or potential strategies to address them.

Lack of External Validation: The results are based on a self-built dataset, and the section does not mention whether external validation or generalization to other datasets was attempted. This raises questions about the model's robustness beyond the training data.

I am a bit concerned about Figure 4 (and that makes me skeptical about the results). 1) The band shows signals of different quality across the channels; 2) How are the channels presented, it is not clear which plot represents which channel

easy to read

Author Response

Please see the attachment.For convenience, I have attached the revised manuscript to the Response Letter.Thank you very much for waiting for my reply.

Round 2

Reviewer 1 Report

Although I might not agree with all the answers given by the authors, they have made most of the important changes. Accordingly, to me, the paper is much better now and I see no problems for publishing it.

Reviewer 2 Report

I thank the Authors for addressing all my concerns. The paper is strongly improved and, in my opinion, it is suitable for publication in the presrnt form.

Reviewer 3 Report

the revisions seem OK. Not sure what the authors mean with Chapter 4. 

NA